# Improving Fairness in AI-Powered Recruitment: An Interpretable Resume Screening System

**Natalia A. Agapova**
Innopolis University
Innopolis, Russia
`n.agapova@innopolis.university`

**Rustam A. Lukmanov**
Research Center of Artificial Intelligence
Innopolis University
Innopolis, Russia
`r.lukmanov@innopolis.university`

## ABSTRACT

Modern automated resume screening systems are typically based on neural text classification models that encode a resume as a feature representation $x \in \mathcal{X}$ and predict a discrete label corresponding to candidate category, suitability level, or job role. Such models commonly produce class logits $\mathbf{z}_\theta(x)$ parameterized by $\theta$, which are converted into class probabilities via the softmax function

$$p_\theta(y = c \mid x) = \frac{\exp(z_{\theta,c}(x))}{\sum_{k=1}^{C_0} \exp(z_{\theta,k}(x))},$$

where $C_0$ denotes the number of target classes in the baseline classification system. These models are typically trained using cross-entropy loss and deployed as the first stage of automated candidate filtering.

Despite their effectiveness, resume classifiers may encode implicit bias through correlations between predictions and non-job-related or proxy textual features. To study this effect, we analyze feature influence using Integrated Gradients, which assign an attribution score to each input feature:

$$\text{IG}_i(x) = (x_i - x_i') \int_0^1 \frac{\partial f_\theta(x' + \alpha(x - x'))}{\partial x_i} \, d\alpha,$$

where $x_i$ denotes the $i$-th input feature, $x'$ is a baseline representation, and $f_\theta$ is a scalar model output such as a class logit or score. This analysis reveals systematic dependencies on features that should be irrelevant to candidate evaluation.

Building on these observations, we evaluate multiple debiasing techniques and propose an interpretability-guided approach to bias mitigation. The model is trained by minimizing a composite objective

$$\mathcal{L} = \mathcal{L}_{cls} + \lambda \, \mathcal{R}_{attr},$$

where $\mathcal{L}_{cls}$ denotes classification loss and $\mathcal{R}_{attr}$ penalizes model reliance on proxy features identified through attribution analysis. This formulation allows explainable analysis to guide the development of fairer resume screening models.

## 1 INTRODUCTION

The adoption of machine learning in Human Resources (HR) has grown rapidly, with automated systems now screening millions of resumes annually (Raghavan et al., 2019). While these systems promise efficiency and consistency, they risk encoding and amplifying societal biases present in historical hiring data.

Bias in automated hiring systems can manifest through multiple protected attributes: gender, age, ethnicity, socioeconomic background, and geographic location. Such biases lead to systematic disadvantages for certain candidate groups—not due to their qualifications, but due to spurious correlations learned by the model. For instance, models may learn to associate certain vocabulary patterns,

institutional names, or regional expressions with hiring outcomes, even when these features carry no legitimate signal about candidate suitability.

In this paper, we present a comprehensive framework for analyzing and mitigating bias in BERT-based resume classification models. We examine multiple potential sources of bias, including geographic location, gender, and age. Our analysis reveals that **geographic bias** (across city groups) exhibits the strongest measurable disparities in our dataset, making it the primary focus of our debiasing efforts. However, the proposed methodology generalizes to any categorical sensitive attribute.

Our contributions are:

1. **Multi-attribute Bias Analysis**: We analyze bias across geographic, gender, and age attributes using True Positive Rate (TPR) gaps, identifying city groups as the dominant source of measurable disparities.

2. **Robust Fairness Metrics**: We introduce a filtering methodology that excludes statistically unreliable subgroups (support $< 30$), revealing that naive fairness metrics can substantially overestimate bias due to small-sample noise.

3. **Interpretability-Driven Bias Detection**: We apply Integrated Gradients to the baseline model to identify specific tokens (city names, age-related terms) that systematically influence predictions, providing evidence of encoded bias before any mitigation.

4. **Comparative Debiasing Study**: We systematically evaluate seven debiasing techniques spanning in-processing methods (GroupDRO, Focal Loss, Label Smoothing, Adversarial) and attribution-based methods (Data Scrubbing, Attention Regularization).

## 2 RELATED WORK

### 2.1 FAIRNESS IN MACHINE LEARNING

Algorithmic fairness has emerged as a critical research area, with multiple competing definitions proposed. Dwork et al. (2012) introduced *fairness through awareness*, arguing that similar individuals should receive similar outcomes. Hardt et al. (2016) formalized *equalized odds*, requiring equal true positive and false positive rates across demographic groups. Chouldechova (2017) demonstrated inherent tensions between calibration and equalized odds, showing that satisfying both simultaneously is often impossible. Caton & Haas (2024) provide a comprehensive survey of fairness metrics and their trade-offs.

### 2.2 BIAS IN HR AND RECRUITMENT SYSTEMS

Automated hiring systems present unique fairness challenges. Raghavan et al. (2019) evaluated commercial resume screening tools and found significant disparities in outcomes across demographic groups. Köchling & Wehner (2020) conducted a systematic review of algorithmic discrimination in HR, identifying data bias, algorithmic bias, and deployment bias as key sources. Chen (2023) examined ethical implications of AI-enabled recruitment, highlighting the tension between efficiency gains and fairness concerns. Deshpande et al. (2020) addressed demographic bias in resume filtering using adversarial debiasing, focusing primarily on gender attributes.

### 2.3 DEBIASING METHODS

Bias mitigation techniques span three stages: pre-processing, in-processing, and post-processing (Hort et al., 2023). Pre-processing methods modify training data through reweighting or resampling. In-processing methods incorporate fairness constraints into the learning objective. Hashimoto et al. (2018) proposed Group Distributionally Robust Optimization (GroupDRO), which optimizes for worst-case group performance:

$$\min_{\theta} \max_{g \in \mathcal{G}} \mathbb{E}_{(x,y) \sim P_g}[\mathcal{L}(f_\theta(x), y)] \tag{1}$$

where $\mathcal{G}$ represents demographic groups. Zhang et al. (2018) introduced adversarial debiasing, training a classifier to be uninformative about protected attributes.

## 2.4 Interpretability for Fairness

Explainable AI (XAI) methods can reveal sources of bias in model predictions. Sundararajan et al. (2017) introduced Integrated Gradients, providing axiomatic guarantees for feature attribution. We use this technique to identify tokens contributing to potentially biased predictions, enabling both diagnosis and targeted mitigation.

**Gap in literature.** While prior work has addressed gender and racial bias in resume screening, comprehensive multi-attribute analysis with robust evaluation metrics remains limited, particularly in non-English contexts, and a definitive solution to model bias remains elusive.

# 3 Problem Setup

## 3.1 Task Definition

We formulate resume classification as a multi-class classification problem. Given a resume text $x \in \mathcal{X}$, we predict the professional category $y \in \mathcal{Y} = \{1, \dots, K\}$ where $K = 9$ categories. Each resume is associated with multiple sensitive attributes: geographic location $g_{city} \in \mathcal{G}_{city}$, gender $g_{gender} \in \{\text{Male}, \text{Female}\}$, and age group $g_{age} \in \mathcal{G}_{age}$.

## 3.2 Dataset

We use a dataset of resumes from the leading job portal in Russia. The dataset comprises 27,550 samples split into training (16,530), validation (5,510), and test (5,510) sets. Resumes span 41 city groups and 7 age categories.

The 9 professional categories are: *backend_general_dev*, *web_frontend*, *sysadmin_devops_network*, *project_product*, *sales_account*, *tech_support_helpdesk*, *it_governance_leadership*, *technical_specialized*, and *generic_it_ops*.

## 3.3 Attribute Distribution Analysis

Before selecting which sensitive attribute to focus on for debiasing, we analyzed the distribution of each attribute:

- **City groups**: 41 unique cities with severe imbalance. Moscow and Saint Petersburg comprise over 40% of samples, while many cities have fewer than 50 samples. This creates substantial subgroup variance.
- **Gender**: Binary distribution (Male/Female) with approximately 70%/30% split. More balanced but fewer distinct groups.
- **Age**: 7 groups ($<18$, 18–21, 22–25, 26–35, 36–50, 50+, Unknown). Moderate imbalance with concentration in 26–35 range.

Preliminary TPR gap analysis revealed that **city groups exhibit the largest measurable disparities** (robust worst-case gap of 32.9% vs. 18.2% for age and 12.4% for gender). We therefore focus our debiasing experiments on city groups while demonstrating the generalizability of our methodology.

# 4 Methodology

## 4.1 Baseline Model

We fine-tune BERT-base-uncased (Devlin et al., 2018) for sequence classification. Input resumes are tokenized with maximum length 128 and classified using a linear head:

$$\hat{y} = \text{softmax}(W \cdot h_{[CLS]} + b) \tag{2}$$

where $h_{[CLS]} \in \mathbb{R}^{768}$ is the BERT [CLS] token representation.

## 4.2 Fairness Metrics

We evaluate fairness using True Positive Rate (TPR) gaps across demographic groups. For a class $c$ and group $g$:

$$\text{TPR}_{c,g} = P(\hat{y} = c | y = c, G = g) \tag{3}$$

The TPR gap measures disparity:

$$\Delta\text{TPR}_c = \max_{g \in \mathcal{G}} \text{TPR}_{c,g} - \min_{g \in \mathcal{G}} \text{TPR}_{c,g} \tag{4}$$

We report **worst-case** gap ($\max_c \Delta\text{TPR}_c$) and **macro-average** gap.

## 4.3 Robust Fairness Evaluation

Small subgroups produce unreliable TPR estimates. We filter group-class combinations with support below $\tau = 30$:

$$\Delta\text{TPR}_c^{\text{robust}} = \max_{g:n_{c,g} \geq \tau} \text{TPR}_{c,g} - \min_{g:n_{c,g} \geq \tau} \text{TPR}_{c,g} \tag{5}$$

## 4.4 Bias Detection via Integrated Gradients

Before applying debiasing methods, we first verify that the baseline model exhibits bias by analyzing which input features influence its predictions. We use Integrated Gradients (Sundararajan et al., 2017), which assigns an attribution score to each input token:

$$\text{IG}_i(x) = (x_i - x_i') \times \int_{\alpha=0}^{1} \frac{\partial F(x' + \alpha(x - x'))}{\partial x_i} d\alpha \tag{6}$$

where $x_i$ is the embedding of the $i$-th token, $x'$ is a baseline (zero embedding), and $F$ is the model's output logit for the predicted class. This method satisfies key axioms including completeness (attributions sum to the prediction difference) and sensitivity (relevant features receive non-zero attribution).

We compute attributions for representative samples across demographic subgroups, selecting examples that illustrate both correct classifications and systematic errors, and analyze tokens related to sensitive attributes: city names and age-related terms.

### 4.4.1 Geographic Bias in Baseline Model

Figure 1 shows the mean attribution scores for city name tokens across professional categories. The results reveal systematic geographic bias:

- **Moscow** has strong *negative* attribution ($-0.164$) for *backend_general_dev*, suggesting the model associates Moscow-based candidates with other professions.
- **Saint Petersburg** shows strong *positive* attribution ($+0.159$) for *sales_account*, indicating geographic stereotyping.
- Regional cities (Ekaterinburg, Novosibirsk, Chelyabinsk) consistently receive positive attribution for *sysadmin_devops_network* ($+0.038$ to $+0.082$).

These patterns demonstrate that the model uses geographic information—which should be irrelevant to professional qualification—as a predictive signal.

Our analysis flagged multiple bias indicators: the token "москва" (Moscow) received inconsistent attributions across samples—negative ($-0.164$) for age 22–25 errors but positive ($+0.197$) for backend misclassifications. Similarly, regional cities like "магнитогорск" (Magnitogorsk) and "томская" (Tomsk) showed systematic positive attribution, confirming geographic signal leakage.

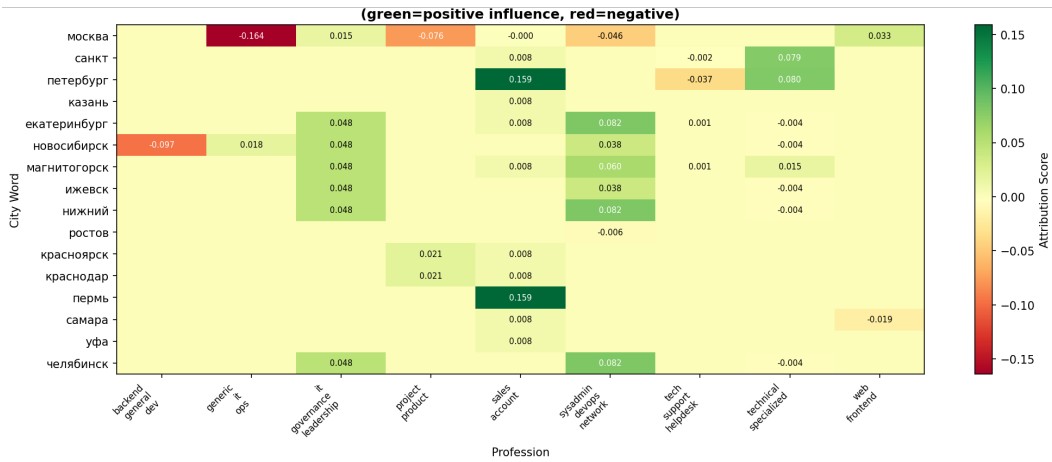

Figure 1: Integrated Gradients attribution scores for city name tokens across professional categories in the baseline model. Green indicates positive influence toward the profession; red indicates negative influence. The model exhibits systematic geographic bias, using city names as predictive features.

### 4.4.2 AGE-RELATED BIAS IN BASELINE MODEL

Figure 2 presents attribution scores for age-related vocabulary. Key findings:

- The token "пенсионер" (pensioner) has a strong negative attribution ($-0.097$) for *backend_general_dev* but positive attribution ($+0.074$) for *it_governance_leadership*, reflecting age stereotypes about technical vs. managerial roles.
- The token "студент" (student) receives positive attribution ($+0.092$) for *sysadmin_devops_network* but negative attribution ($-0.014$) for *tech_support_helpdesk*.
- Age numbers ("50", "60") show differential patterns: older ages correlate negatively with technical roles and positively with governance positions.

These interpretability findings confirm that the baseline model learns spurious correlations between protected attributes and professional categories, motivating the debiasing methods described below.

We did not include a gender attribution heatmap because no explicit gender-indicative tokens were found in the data; gender bias may be encoded through indirect proxies (surnames, job titles) not captured by token-level analysis.

Additional per-sample attribution examples are provided in Appendix A.

### 4.5 DEBIASING METHODS

Based on the bias patterns identified through Integrated Gradients analysis, we evaluate seven debiasing techniques:

### 4.5.1 INVERSE SQUARE-ROOT REWEIGHTING

Sample weights inversely proportional to group frequency:

$$w_i = \frac{1}{\sqrt{n_{g_i}}} \cdot \frac{1}{Z} \tag{7}$$

### 4.5.2 GROUPDRO

Optimizes for worst-case group performance with dynamic group weights:

$$q_g^{(t+1)} \propto q_g^{(t)} \cdot \exp(\eta \cdot \mathcal{L}_g^{(t)}) \tag{8}$$

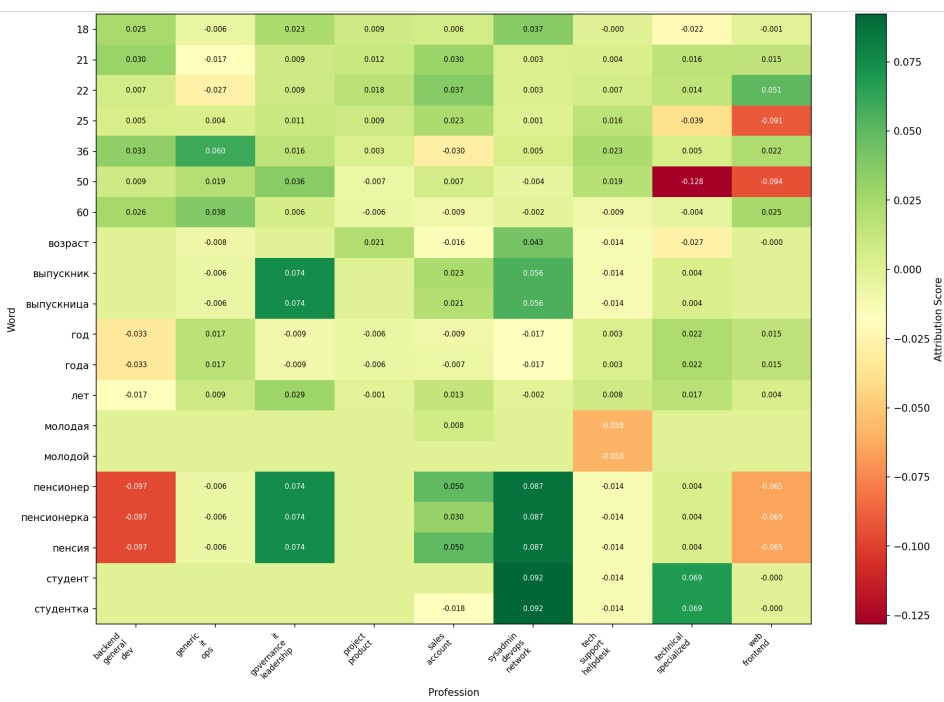

Figure 2: Attribution scores for age-related words across professions. The baseline model encodes age stereotypes: "pensioner" negatively influences backend development predictions while positively influencing IT governance.

### 4.5.3 FOCAL LOSS

Down-weights well-classified examples:

$$\mathcal{L}_{FL} = -\alpha_t(1-p_t)^\gamma \log(p_t) \tag{9}$$

### 4.5.4 LABEL SMOOTHING

Distributes probability mass to non-target classes:

$$y'_c = (1-\epsilon) \cdot y_c + \frac{\epsilon}{K} \tag{10}$$

### 4.5.5 ADVERSARIAL DEBIASING

Trains a classifier while preventing encoding of protected attributes via gradient reversal:

$$\mathcal{L} = \mathcal{L}_{cls} - \lambda\mathcal{L}_{adv} \tag{11}$$

### 4.5.6 DATA SCRUBBING (FAIRNESS THROUGH BLINDNESS)

Directly motivated by our Integrated Gradients analysis, we mask the sensitive tokens identified in Section 4.4. Using the attribution heatmaps (Figures 1 and 2), we construct a set $\mathcal{S}$ of tokens with high attribution magnitude on sensitive attributes:

$$x'_i = \begin{cases} \texttt{[MASK]} & \text{if } t_i \in \mathcal{S} \\ x_i & \text{otherwise} \end{cases} \tag{12}$$

The set $\mathcal{S}$ includes city names (Moscow, Saint Petersburg, etc.) and age-related terms (pensioner, student, age numbers).

Table 1: Comparison of debiasing methods on test set. Robust metrics use $\tau = 30$. TPR gaps reported as decimals (lower is better for fairness). Best results in bold; worst marked with †.

| Model | Acc | F1 | Worst Gap | Macro Gap |
|---|---|---|---|---|
| Baseline (BERT) | **0.609** | 0.621 | 0.329 | 0.116 |
| *In-processing methods* | | | | |
| + Label Smoothing $\epsilon$=0.1 | 0.596 | 0.608 | 0.329 | 0.140 |
| + Focal Loss $\gamma$=2 | 0.593 | 0.613 | 0.329 | 0.136 |
| + Adversarial $\alpha$=0.5 | 0.574 | 0.594 | 0.429† | 0.132 |
| + GroupDRO $\eta$=0.3 | 0.552 | 0.572 | 0.282 | 0.135 |
| + GroupDRO $\eta$=0.1 | 0.548 | 0.553 | **0.265** | 0.113 |
| *Attribution-based methods* | | | | |
| + Data Scrubbing | 0.594 | 0.611 | 0.300 | **0.112** |
| + Attention Reg $\lambda$=0.1 | 0.583 | 0.601 | 0.335 | 0.116 |

### 4.5.7 ATTENTION REGULARIZATION

We add a penalty term that discourages the model from attending to sensitive tokens:

$$\mathcal{R}_{attr} = \frac{1}{|\mathcal{B}|} \sum_{x \in \mathcal{B}} \sum_{i:t_i \in \mathcal{S}} a_i^{[CLS]} \tag{13}$$

where $a_i^{[CLS]}$ is the average attention weight from the [CLS] token to position $i$ across all layers and heads.

## 5 EXPERIMENTS

### 5.1 EXPERIMENTAL SETUP

All models use BERT-base-uncased with learning rate $2 \times 10^{-5}$, batch size 8, max sequence length 128, and 2 training epochs.

### 5.2 RESULTS

Key findings:

**GroupDRO achieves the strongest fairness improvement.** With $\eta = 0.1$, worst-case TPR gap decreases from 32.9% to 26.5%. However, this comes at a significant accuracy cost (6.1%). The $\eta = 0.3$ variant offers a middle ground with 28.2% worst gap and 5.7% accuracy loss.

**Data Scrubbing provides the best accuracy-fairness trade-off.** By masking the sensitive tokens identified through our Integrated Gradients analysis (Section 4.4), this method reduces worst-case gap to 30.0% and achieves the best macro gap (11.2%) while losing only 1.5% accuracy. This demonstrates the practical value of interpretability-guided debiasing.

**Adversarial debiasing fails catastrophically.** Rather than reducing bias, it increases worst-case gap to 42.9%, much worse than the baseline. We hypothesize that with 41 city groups, the adversarial head becomes too difficult to train, leading to unstable gradients.

**Focal Loss and Label Smoothing provide minimal fairness benefits** while slightly degrading accuracy. These methods primarily address class imbalance rather than group-level disparities.

**Attention Regularization shows limited effectiveness.** Despite penalizing attention on sensitive tokens, this method slightly worsens fairness (33.5% vs 32.9% baseline), suggesting that attention-level intervention alone is insufficient without representation-level changes.

## 6 DISCUSSION

### 6.1 WHY GEOGRAPHIC BIAS DOMINATES

Our multi-attribute analysis revealed that city groups exhibit larger TPR gaps than gender or age. We hypothesize three contributing factors:

1. **Vocabulary variation**: Regional vocabulary patterns, company names, and institutional references create strong textual signals correlated with geography.

2. **Data imbalance**: The 41 city groups create a much more fragmented attribute space than binary gender, amplifying variance.

3. **Historical hiring patterns**: Geographic concentration of certain industries may create legitimate correlations that nevertheless constitute unfair treatment.

### 6.2 THE VALUE OF INTERPRETABILITY

Our Integrated Gradients analysis (Section 4.4) served two purposes: (1) confirming that the baseline model encodes bias through specific token attributions, and (2) directly informing the Data Scrubbing method by identifying which tokens to mask. This interpretability-guided approach achieved competitive fairness improvements with minimal accuracy loss, demonstrating that understanding *how* a model is biased enables more targeted mitigation.

### 6.3 LIMITATIONS

- Our dataset is limited to Russian IT resumes; generalization requires validation.

- Post-processing calibration methods were not explored.

- Intersectional analysis (e.g., female candidates from small cities) remains for future work.

## 7 CONCLUSION

We presented a comprehensive framework for analyzing and mitigating bias in resume classification. Using Integrated Gradients, we first demonstrated that the baseline BERT model encodes systematic bias through city names and age-related vocabulary. Our analysis of multiple sensitive attributes revealed that geographic bias exhibits the strongest measurable disparities.

Our comparative study of seven debiasing techniques yields several key insights:

- **GroupDRO** achieves the best fairness improvement but at significant accuracy cost.

- **Data Scrubbing**, guided by interpretability analysis, offers the best practical trade-off, improving fairness while losing only 1.5% accuracy.

- **Adversarial debiasing** fails with many demographic groups (41 cities), highlighting the importance of method selection based on attribute cardinality.

- **Attention regularization** alone is insufficient; representation-level or input-level changes are more effective.

Future work should explore intersectional analysis, post-processing calibration, and deployment in real-world hiring pipelines.

## ACKNOWLEDGMENTS

This work was supported by the Academy of Sciences of the Republic of Tatarstan, Contract No. 254/2024-PD

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

# A  ADDITIONAL ATTRIBUTION EXAMPLES

This appendix presents additional IG analysis results supporting the findings in Section 4.4.

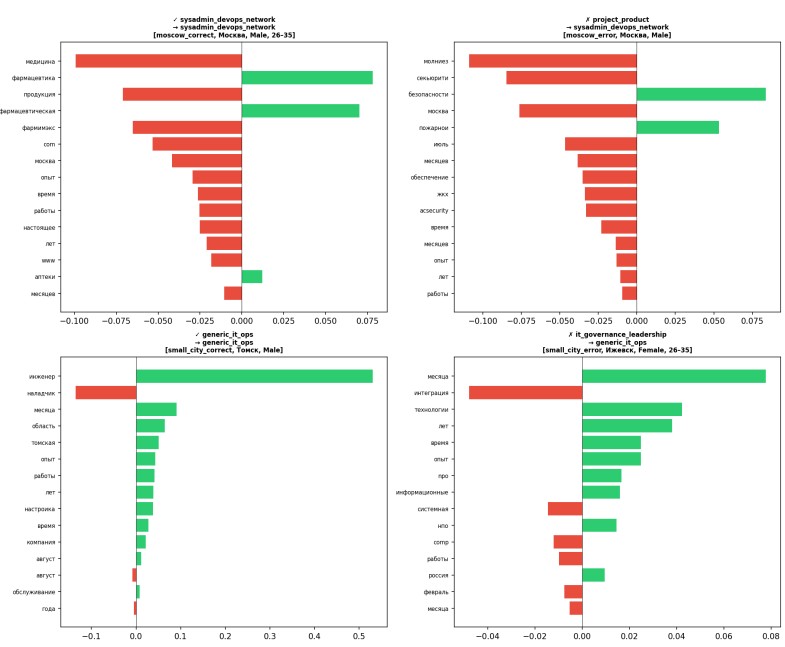

Figure 3: Token attributions: Large vs. small cities.

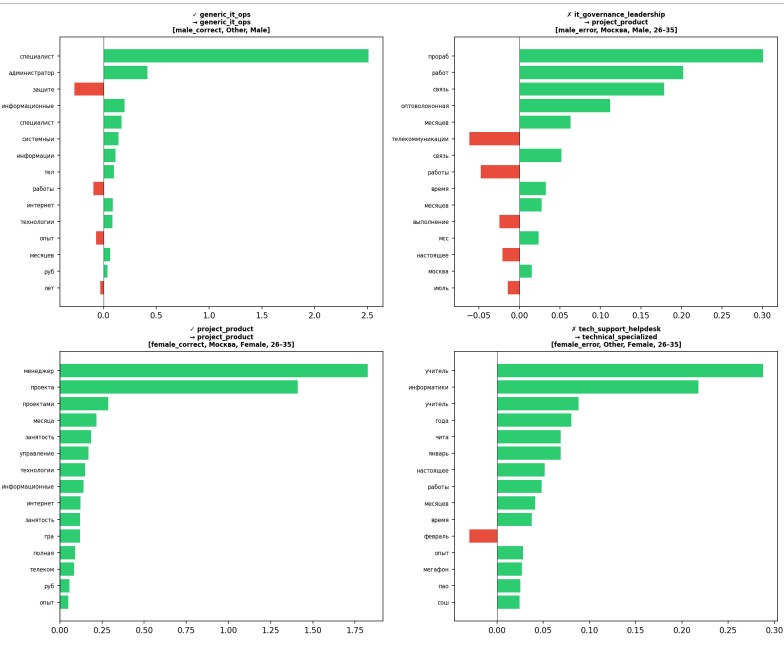

Figure 4: Token attributions by gender.

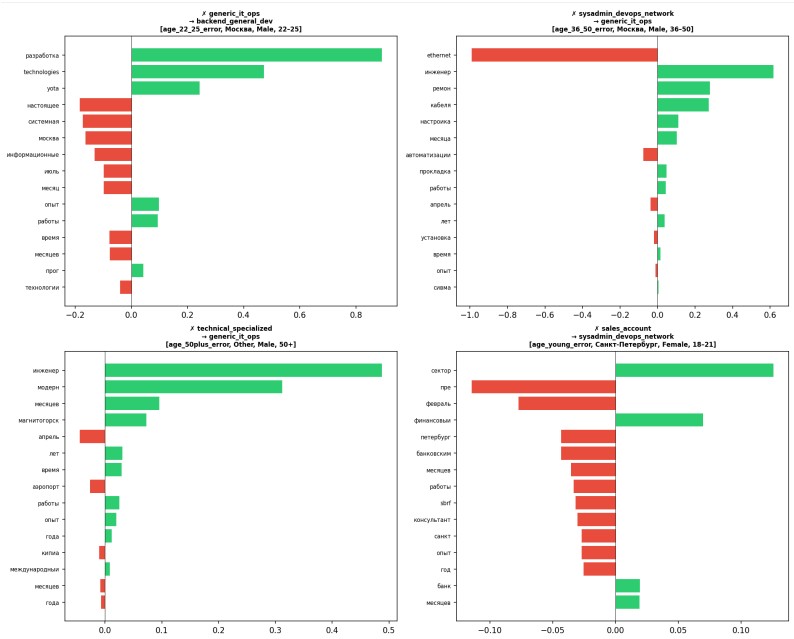

Figure 5: Token attributions across age groups.

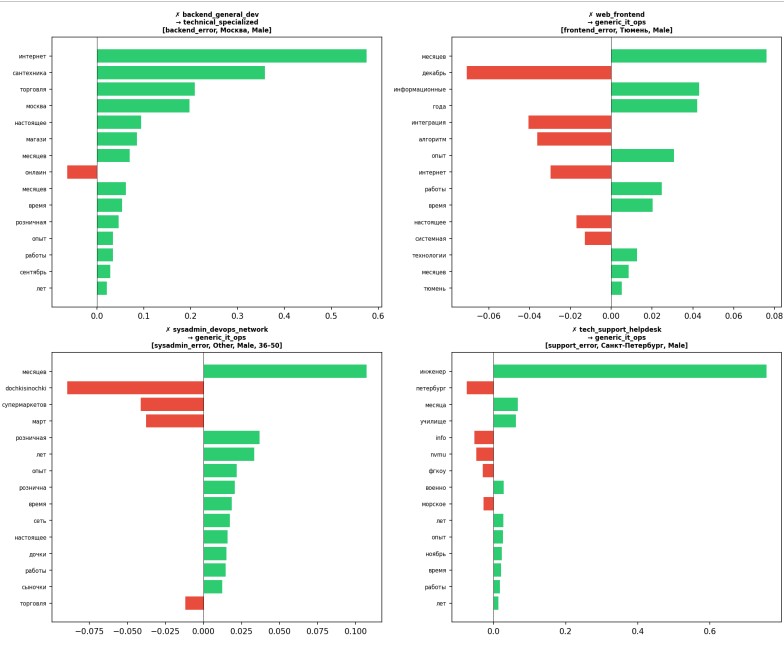

Figure 6: Attribution analysis of profession misclassifications.

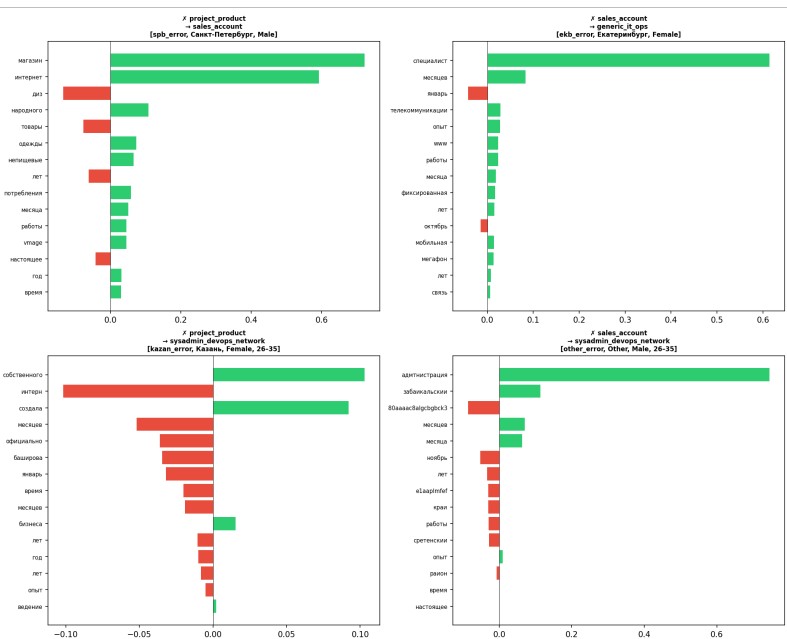

Figure 7: Attribution analysis for regional cities.

