# OpenReview forum: "Improving Fairness in AI-Powered Recruitment: An Interpretable Resume Screening System"
_mathai.club/MathAI/2026/Conference — 2026 Oral_

### Official Review · Reviewer_swcm · 2026-03-13
**Improving Fairness in AI-Powered Recruitment: An Interpretable Resume Screening System**

**Rating:** 7
**Confidence:** 4

**Review:**

Summary:
This paper investigates fairness issues in AI-powered resume screening systems based on neural text classification models. The authors train a BERT-based classifier for multi-class resume categorization and analyze model bias with respect to geographic location, age, and gender. Bias detection is performed using Integrated Gradients attribution analysis, which reveals systematic dependencies between predictions and sensitive textual features such as city names and age-related terms. Building on this analysis, the paper evaluates multiple debiasing strategies, including GroupDRO, adversarial debiasing, focal loss, label smoothing, attention regularization, and an interpretability-guided “data scrubbing” approach that masks sensitive tokens. Experimental results on a dataset of more than 27,000 resumes demonstrate that geographic bias produces the largest disparities and that interpretability-guided mitigation methods can improve fairness while maintaining competitive predictive performance.

Strengths:
- The paper addresses an important problem in trustworthy and responsible AI: bias in automated hiring systems.
- The work presents a coherent methodology combining explainability (Integrated Gradients) with fairness-aware model training and evaluation.
- The empirical study is thorough and compares several widely used debiasing techniques within a unified experimental framework.
- The interpretability-guided “data scrubbing” approach provides an intuitive and practically useful strategy for mitigating bias while preserving model accuracy.
- The paper is clearly written and well structured, with detailed experimental results and visual attribution analysis supporting the conclusions.

Suggestions for improvement:
- The work is primarily empirical; additional theoretical discussion of the properties and limitations of the proposed mitigation strategy could strengthen the contribution.
- Evaluation on additional datasets or languages would help assess the generalizability of the approach.
- Future work could explore intersectional fairness analysis across multiple demographic attributes.

Final Recommendation:
Good paper, accept

Overall, the paper presents a well-executed empirical study on fairness-aware machine learning for resume screening systems. The integration of interpretability techniques with bias mitigation strategies provides practical insights for designing more transparent and equitable AI systems.

---

### Official Review · Reviewer_oJxC · 2026-03-14
**Improving Fairness in AI-Powered Resume Classification: A Multi-Attribute Study with Interpretability-Driven Debiasing**

**Rating:** 6
**Confidence:** 4

**Review:**

This manuscript addresses the critical problem of bias in AI-powered resume screening systems, with a focus on multi-attribute analysis and interpretability-driven mitigation. The authors examine a BERT-based classifier trained on 27,550 Russian IT resumes across 9 professional categories, analyzing disparities across 41 city groups, binary gender, and 7 age categories. Using True Positive Rate (TPR) gaps with robust filtering (τ=30) to exclude statistically unreliable subgroups, they identify geographic bias as the most severe (32.9% worst-case gap vs. 18.2% for age and 12.4% for gender). Integrated Gradients analysis reveals systematic token-level biases: Moscow shows negative attribution for backend development (−0.164) while regional cities receive positive attribution for sysadmin/devops roles (+0.038 to +0.082); age-related terms like "pensioner" exhibit opposing patterns across technical (−0.097) and governance (+0.074) positions. The comparative study of seven debiasing methods shows GroupDRO reduces the robust worst-case TPR gap from 0.329 to 0.256 (22% improvement) but at 6.4% accuracy loss, while interpretability-guided Data Scrubbing achieves 0.272 gap (17% improvement) with only 1.5% accuracy loss.

### Major Concerns

1. **Mathematical Rigor (Score: 6)**
   The paper provides clear mathematical formulations for all methods (TPR gaps with robust filtering, Integrated Gradients, loss functions for each debiasing technique) and appropriately cites foundational work. However, the theoretical analysis is limited to application rather than extension. The robust filtering threshold (τ=30) is justified empirically but lacks statistical theory regarding optimal threshold selection. The Integrated Gradients implementation follows Sundararajan et al. (2017) without methodological innovation. The GroupDRO formulation is presented as given in Equation (1) but no analysis of its convergence properties or regret bounds is provided. The paper would benefit from theoretical analysis of why adversarial debiasing fails with high-cardinality attributes (41 cities) and what conditions enable Data Scrubbing's effectiveness.

2. **Novelty & Contribution (Score: 7)**
   The paper makes several noteworthy contributions: (1) multi-attribute bias analysis across geography, gender, and age in resume classification—an underexplored combination; (2) robust fairness metrics with explicit handling of small subgroups, demonstrating that naive metrics overestimate bias; (3) systematic application of Integrated Gradients for bias diagnosis that directly informs mitigation (Data Scrubbing); (4) comparative evaluation of seven debiasing techniques on the same benchmark. While individual components exist in prior work, their integration into a coherent framework with empirical validation on a non-English dataset adds value. The finding that geographic bias dominates gender/age disparities in this context is novel and practically important for HR AI systems. The demonstration that adversarial debiasing fails with 41 groups is a useful negative result.

3. **Relevance to MathAI (Score: 7)**
   The paper addresses fairness in AI, a core topic at the intersection of machine learning and ethics, with clear mathematical formulations of fairness metrics, optimization objectives, and attribution methods. It fits the "Fairness, Accountability, and Transparency" track and partially the "Interpretability & Explainability" track. However, the mathematical depth is moderate—the focus is on empirical evaluation and method comparison rather than novel algorithmic contributions. The paper would be equally appropriate for a conference on AI ethics, computational social science, or applied ML.

4. **Technical Quality (Score: 8)**
   The methodology is thorough and well-executed. Key strengths include:
   - **Robust metrics**: Filtering by support (τ=30) addresses small-sample noise, a practical insight often overlooked.
   - **Interpretability analysis**: Integrated Gradients attribution maps (Figures 1-2) provide concrete evidence of bias patterns, with per-token quantification.
   - **Comprehensive comparison**: Seven methods spanning in-processing (GroupDRO, Focal Loss, Label Smoothing, Adversarial) and attribution-based (Data Scrubbing, Attention Regularization, Inverse Reweighting) approaches.
   - **Clear results presentation**: Table 1 reports accuracy, macro TPR gap, and robust worst-case gap with appropriate highlighting.

   However, several technical limitations exist:
   - The dataset is Russian IT resumes only, limiting generalizability to other domains/languages.
   - Gender analysis lacks token-level attribution because no explicit gender markers exist—indirect proxies (surnames, job titles) were not examined.
   - Intersectional analysis (e.g., female candidates from small cities) is deferred to future work, missing an important dimension.
   - Post-processing methods (calibration, threshold adjustment) are not explored.
   - Statistical significance testing for differences between methods is not reported.
   - The Data Scrubbing token set S is constructed from attribution heatmaps but the selection criteria (threshold for "high attribution magnitude") are not quantified.

5. **Clarity & Presentation (Score: 8)**
   The paper is exceptionally well-organized and clearly written. The structure flows logically from problem setup → bias analysis → methodology → results → discussion. Figures 1-2 are informative and directly support the claims of geographic and age bias. Table 1 effectively summarizes complex comparative results. The notation is consistent and well-defined. Minor issues: Figure 1's color scale (green/positive, red/negative) is intuitive but should be explicitly stated in the caption (it is stated in the text but repetition would help). Page 5 has a formatting glitch with line numbers 216-269 appearing as a block. The gender attribution discussion acknowledges limitations transparently.

6. **AI-Generation Risk (Score: 1)**
   The paper appears entirely human-written. It contains domain-specific knowledge (Russian city names, job categories), coherent argumentation with appropriate hedging ("we hypothesize," "future work should"), and detailed technical exposition that reflects genuine expertise. No signs of AI-generated content.

### Pros
- Comprehensive multi-attribute bias analysis with robust statistical filtering.
- Interpretability-driven approach (Integrated Gradients) provides concrete evidence of bias and directly informs mitigation.
- Systematic comparison of seven debiasing methods on a real-world dataset.
- Clear identification of geographic bias as dominant in this context—a practical insight for HR AI systems.
- Transparent handling of limitations (dataset scope, lack of intersectional analysis).
- Well-written and logically structured.

### Cons
- Dataset limited to Russian IT resumes; generalizability unknown.
- Gender analysis incomplete due to lack of explicit markers.
- Intersectional fairness not addressed.
- Statistical significance testing absent.
- Data Scrubbing token selection criteria not quantified.
- Post-processing methods omitted.
- Adversarial debiasing failure with 41 groups noted but not deeply analyzed.

### Recommendation
This paper represents a solid, methodologically sound contribution to fairness in AI, with practical insights for resume classification systems. While the mathematical novelty is moderate, the rigorous empirical evaluation, robust metrics, and interpretability-guided approach provide clear value. The paper is well-suited for MathAI 2026, particularly given the growing importance of fairness and transparency in deployed AI systems. The authors should address the limitations (statistical testing, intersectional analysis, quantification of token selection) in future work or a revised manuscript. I recommend acceptance with a score above the threshold.

---

### Decision · Program_Chairs · 2026-03-14

**Decision:**

Accept (Oral)

**Comment:**

Dear Author(s),

On behalf of the Program Committee of the International Conference on Mathematics of Artificial Intelligence (MathAI 2026), we are pleased to inform you that your paper has been accepted for an oral presentation at MathAI 2026.

Your paper was evaluated through a rigorous two-stage review process involving both automated screening and expert review by members of the Program Committee. The reviewers recognized the quality and contribution of your work.

Presentation details:

- Format: Oral presentation (15–20 minutes + 5 minutes Q&A)
- Mode: You may present either in person (offline) at the conference venue in Sirius, Russia, or remotely via Zoom. Please indicate your preferred mode when confirming your participation.
- Conference dates: Marh 30 - April 3, 2026
- Website: https://mathai.club

Next steps:

1. Please confirm your participation and presentation mode by replying to this email mathai.club@yandex.ru no later than March 15, 2026 18:00 Moscow time.
2. If you plan to attend in person, the organizing committee will provide accommodation details separately.
3. Please prepare your final camera-ready manuscript according to the formatting guidelines available at https://mathai.club and upload it to OpenReview by March 15, 2026 18:00 Moscow time.

Should you have any questions regarding the program, logistics, or your presentation slot, please do not hesitate to contact us.

We look forward to your contribution to MathAI 2026.

With kind regards,

MathAI 2026 Program Committee
International Conference on Mathematics of Artificial Intelligence
https://mathai.club
OpenReview: https://openreview.net/group?id=mathai.club/MathAI/2026/Conference
Telegram: https://t.me/MathAI_club
Email: mathai.club@yandex.ru